# Provider perspectives on empirical antibiotic treatment for tuberculosis-like symptoms in South Africa's private general practice sector: A qualitative study in two cities

**Jeremiah Chikovore**[1]*, **Jody Boffa**[2], **Sizulu Moyo**[3,4], **Angela Mak**[5,6], **Zimasa Gavu**[1], **Angela Salomon**[7], **Madhukar Pai**[7,8], **Amrita Daftary**[6,9,10]

**1** Human Sciences Research Council (HSRC), Public Health, Society and Belonging (PHSB) Division, Durban, South Africa, **2** The Aurum Institute, Johannesburg, South Africa, **3** HSRC, PHSB Division, Cape Town, South Africa, **4** University of Cape Town, Cape Town, South Africa, **5** Cumming School of Medicine, University of Calgary, Calgary, Canada, **6** Dahdaleh Institute of Global Health Research, York University, Toronto, Canada, **7** McGill University Health Centre, McGill University, Montréal, Canada, **8** Department of Global and Public Health and McGill International TB Centre, McGill University, Montréal, Canada, **9** School of Global Health, York University, Toronto, Canada, **10** Centre for the AIDS Programme of Research in South Africa (CAPRISA), University of KwaZulu-Natal, Durban, South Africa

* jchikovore@hsrc.ac.za

## Abstract

While tuberculosis (TB) in South Africa is commonly treated in the public sector, some clients first seek care in the private sector. Research has demonstrated that private general practitioners (GPs) perform less well than do public sector care providers in TB testing and drug-dispensing practices. We aimed to describe GPs' decision-making practices related to empiric antibiotic treatment when presented with symptoms that may be related to TB, to inform potential interventions. Within a larger study on private sector TB management, we qualitatively interviewed 30 purposively selected GPs, who varied by gender, age, practice community, and how they managed TB and HIV in the parent study. Data were analysed through coding and constant comparison. GPs acknowledged the common use of broad-spectrum antibiotics for respiratory symptoms, driven by experience treating presumed bacterial infections and by a desire to rule out other causes before referring clients for potentially inconvenient TB tests in the private or public sector. Management decisions were susceptible to perceived or expressed pressure from clients, who may expect on-the-spot treatment. Additionally, GPs indicated using antibiotics to mitigate financial strain on economically vulnerable clients. Empirical antibiotic treatment for presentations that may be related to TB in the private sector, which can delay TB diagnosis, could be explained by the absence of accessible and affordable TB and general bacteriologic tests at the point of care, leading GPs to, among others, seek to 'rule out' possible bacterial infection. Potential interventions include increasing the

**Data availability statement:** All data relevant to the study analysis are available within the article. However, the full qualitative data set cannot be shared publicly in order to protect participant anonymity and confidentiality, in accordance with the study's ethics protocols. Ethical approval for the study was granted by the Human Sciences Research Council Research Ethics Committee (HSRC REC) in South Africa (HSRC Ref#: 2/18/10/17) and the McGill University Health Centre (MUHC) Research Ethics Board in Canada (MUHC Ref#: SP SA / 2018-4390). For any inquiries regarding data access, interested parties may contact the Chair of the HSRC REC at research. integrity@hsrc.ac.za or the Co-Chair of the MUHC REB Clinical Trials 2 (CT2) Panel at reb. ct2@muhc.mcgill.ca, citing the relevant study approval number.

**Funding:** This work was supported in its entirety by the Bill & Melinda Gates Foundation [Grant Number OPP1172634]. Under the grant conditions of the Foundation, a Creative Commons Attribution 4.0 Generic License has already been assigned to the Author Accepted Manuscript version that might arise from this submission. The funder had no role in the study design, data collection, data analysis, data interpretation, or writing of the report.

**Competing interests:** The authors have declared that no competing interests exist.

salience of inappropriate antibiotic use, heightening GPs' suspicion index for TB, and linking GPs directly to public sector TB tests for clients.

## Introduction

Globally in 2023, approximately 10.8 million people acquired, and 1.25 million died from the infectious disease, tuberculosis (TB) [1]. With an incidence of 427 per 100,000, South Africa is among the 30 countries with the highest burdens of TB worldwide [1]. The country also has an especially high rate of multidrug-resistant (MDR) TB and HIV-associated TB. MDR-TB is resistant to the most common anti-TB drugs and therefore harder to treat. In 2022, the incidence of TB, DR-TB, and TB among people living with HIV (PLHIV) was estimated at 280 000, 11 000, and 152 000, respectively [2].

Managing TB requires early identification and effective treatment to prevent health sequelae, death, transmission, drug resistance, distress, and economic hardship [3], and this calls for quality healthcare. The World Health Organisation (WHO) recommends that people with pulmonary TB symptoms be tested using nucleic acid amplification tests, e.g., Xpert Ultra [4]; that the positive receive standardised anti-TB treatment (ATT) [5,6], and the negative with prior contact with a person with TB receive preventive treatment. In South Africa, Xpert Ultra, the standard first-line test [7], was rolled out in 2017 and by 2019, 203 testing sites were servicing 4 710 facilities [8]. A TB diagnosis can nevertheless occur and ATT initiated without microbiological confirmation, based on symptoms, chest X-ray results, comorbidities especially HIV, and/or a lack of response to broad-spectrum antibiotics [9]. Among PLHIV, guidance urges routine TB testing, ATT for those found to have active disease, and preventive treatment for those without active TB [10].

The South African health system is two-tiered, with a private well-funded and resourced health system catering for a wealthier but smaller segment, and a less financed public system catering for the poorer but larger segment [11]. As with most high-TB-burden countries, public health systems face equipment and staffing shortages, long waiting times, lack of privacy and courtesy [12], and specifically in South Africa, fragmentation (within the sector, and between the public and private sectors), and poor human resource management and leadership [12–14]. The *Apartheid* legacy and subsequent emergence and consolidation of private healthcare services contributed to weaknesses and inequities currently characterising the public health sector [12,14].

Detection of people with TB in most high-TB-burden countries has typically relied on people with symptoms self-presenting to the health sector – or passive case finding [15,16], whose inadequacies are well described [17]. The First National TB Prevalence Survey in South Africa (2018) established that 14.7% of survey participants reported TB symptoms [18]. In South Africa TB treatment is freely available through the public sector, and non-public providers are required to refer potential TB patients to the public sector [19]. Between a quarter and a third of households in the country

first seek care from private doctors and hospitals for any illness [20,21]. Although many people may thereafter move to the public sector, the private sector continues to provide primary care services to 28–38% of people [22]. A five-country patient pathway analysis also revealed that up to 60% of people with TB began by seeking care in the private sector [23]. Private healthcare providers may offer levels of care that vary from National TB Programme or international guidelines and standards [3], and studies in South Africa have reported greater diagnostic delays and lesser likelihood to order sputum tests among clients who initially visit a private provider [24–26]. In 2022, an estimated 65 578 persons nationally developed TB but were not notified, and treatment coverage was 77% [27].

Thus far, there has been limited attention to the quality of care in South Africa's private health sector, hence little understanding exists of private provider decision-making processes and the challenges faced and strategies used to overcome them [28,29]. This gap may partly arise from expectations or assumptions the sector's extensive resources [30,31] translate to person-centred and high-quality care. A study of health seeking behaviour among low-income patients in South Africa affirmed views among patients of private healthcare services being of greater quality [32]. In a previous study, we used the standardised patient methodology to describe the quality of TB and HIV-associated TB diagnostic care, and quantify the know-do gap (difference between providers' knowledge and/or intentions and the actual observed or described practices) among 212 private GPs practising in two urban areas in South Africa [33,34]. We identified missed opportunities to appropriately investigate or refer for TB and HIV testing [33]. Despite reporting TB-suggestive symptoms, including sometimes producing a confirmed laboratory report, appropriate testing, referral, or treatment were not initiated in more than 50% of interactions. Limited efforts were made to inquire about HIV status. This is despite policy support for TB-HIV service integration [9,35]. Furthermore, a small-scale study in South Africa reported that 42% of HIV clients received antiretroviral therapy (ART) through the private sector [32], while a multicountry global-wide analysis also reported that up to 45% of women and 42% of men reported their most recent testing as done in the private sector [36]. The know-do gap in our parent study was 37% for TB and 18% for HIV [33], suggesting that factors other than knowledge likely affect care practices. Medicines were dispensed or prescribed in almost 90% of interactions, at 3.1 medications per visit, with antibiotics specifically in 77% of interactions. Other common medications included cough syrups (20.7% of all medicines) and analgesics (12.9% of all medicines), which are primarily prescribed to abate symptoms [33,34]. The present qualitative sub-study sought to enhance insights into the contextual dynamics of consultation considerations, and in particular, antibiotic use by private GPs, in the care provision for people with TB-related symptoms.

## Materials and methods

### Ethics statement

Ethics approval was granted by the Human Sciences Research Council Ethics Committee (HSRC Ref#: 2/18/10/17) and the McGill University Ethics Board (MUHC Ref#: 2018--4390). Informed consent was sought and obtained from all participants, and anonymity and confidentiality were maintained throughout the research process. The interviews were audio-recorded with the participants' consent.

### Design, selection of participants, and data collection approach

The parent study took place in eThekwini and Cape Town from August 2018–July 2019, and the GPs were recruited from selected high-TB-burden communities in the two cities. (For the detailed research approach, see [33,34]). Subsequent to unannounced visits from standardised patients in the parent study, a subsample of consenting GPs was recruited and privately interviewed using qualitative methods. GPs were purposively chosen to include those who managed TB and/or HIV according to national guidelines (i.e., referred for a TB test or to the public sector for further care, and/or inquired about HIV status or initiated an HIV test) and those who did not, in the parent study. They were further chosen to achieve variation by age, gender, and location (areas likely to have client profiles from different ethnicities and socioeconomic

groups), to obtain the breadth of dimensions to the topic [37]. We targeted interviewing 30 GPs due to feasibility and an expectation to achieve saturation of relevant themes [38].

Interviews took place at GPs' practices and lasted approximately 30 minutes. The questions were based on an open-ended guide generated from the study goals and leads from preliminary analysis of the larger study. Subsequently, typical of qualitative research approaches, questions were continually reviewed to allow the pursuit of crucial emerging leads [39,40]. (The core guide is included as a supplementary file, S1 Text). In addition to background information about the practice (such as client demographics, challenges commonly encountered as well as positive aspects about the practice, and typical complaints seen), the guide covered the following interrelated topics: i) mechanisms of decision-making around empirical treatment in response to respiratory symptoms; ii) accompanying actions and underlying rationale for presuming or not presuming TB; and iii) consideration, if any, given during consultation to a client's contextual (social, economic, familial) circumstances. Research assistants (n = 5) were trained in qualitative interviewing including nonevaluative approaches to probing. The participants were not informed of evaluations of their actual practices from the main study, although their qualitative interview responses were also analysed in the context of those evaluations (see below).

### Data management and analysis

Preliminary analysis, involving debriefing between lead researchers and research assistants, closely tracked data collection in each city, starting with eThekwini, and enabled immediate data review [40,41] and adjustment of questions to pursue emerging leads or sharpen focus. Audio files were assigned pseudonyms and stored on password-protected accounts, then transcribed and translated verbatim. All hard-copy data were similarly assigned pseudonyms and filed in lockable cabinets accessible only to key staff. Descriptions of the facilities and consent forms were stored separate from the transcripts. A separate protected spreadsheet linked pseudonyms to demographics for analysis purposes. Names that were drawn at the stage of listing were filed separately from the data and files.

Transcripts were checked for accuracy and analysed both inductively and deductively [42,43]. Within a process of data familiarisation (or making broad impressions [44]), the data were initially read and coded [45] in large chunks, relying partly on the study objectives and partly on preliminary analyses from the larger study. Through emergent identification of their properties and/or reference to interview guide sub-questions, the large segments of coded text were further broken down to identify finer dimensions. Relationships between the codes were identified and refined through constant comparison [46]. The text coded with the refined frame was retrieved and critically reviewed in the context of GPs' practices from the larger study, the study settings, and the wider TB epidemiological and policy milieu [47,48]. At this stage, the data were simultaneously being recoded and reconnected in more abstract ways to generate high-level themes, which we present below. The handling and retrieval of data and coded text were facilitated with the use of NVivo qualitative analysis software version 12 (QSR International). JC led the initial coding and analysis and generated preliminary themes. JB and AD then reviewed the codes and preliminary themes based on the larger study's findings, their reading of the transcripts, and theoretical considerations. They generated confirmatory and alternate perspectives which were discussed among the three. The analysis was then shared, debated and refined over repeated consultations with the full team. In addition to this researcher triangulation [40], member checking/ participant validation was done by presenting findings to GPs, whose insights helped consolidate the interpretation of findings.

We present findings as themes supported by illustrative anonymised quotes with limited identifiers (gender, approximate age, and measured practice) to provide context. The study is presented according to COREQ (consolidated criteria for reporting qualitative research) guidance for reporting qualitative research [49].

### Findings

### Overview of GPs and their practices

All GPs approached for qualitative interviews consented: 15 from eThekwini (2 F, 13 M), and 15 from Cape Town (3 F, 12 M). The providers' characteristics are described in Table 1. Their sociodemographic characteristics were similar to those of the

**Table 1. Provider characteristics.**

| Characteristic | Overall n = 30 | eThekwini n = 15 | Cape Town n = 15 |
|---|---|---|---|
| Sex, n (%) | | | |
| Female | 5 (16.7) | 2 (13.3) | 3 (20.0) |
| Male | 25 (83.3) | 13 (86.7) | 12 (80.0) |
| Approximate age, n (%) | | | |
| 30s | 6 (20.0) | 4 (26.7) | 2 (13.3) |
| 40s | 7 (23.3) | 3 (20.0) | 4 (26.7) |
| 50s | 8 (26.7) | 3 (20.0) | 5 (33.3) |
| 60s | 6 (20.0) | 4 (26.7) | 2 (13.3) |
| 70s | 2 (6.7) | 0 (0) | 2 (13.3) |
| 80s | 1 (3.3) | 1 (6.7) | 0 (0) |
| Area, n (%) | | | |
| City | 8 (26.7) | 8 (53.3) | 0 (0.0) |
| Township | 8 (26.7) | 6 (40.0) | 2 (13.3) |
| Suburb | 14 (46.7) | 1 (6.7) | 13 (86.7) |
| Years in practice, median (IQR) | 24 (13-33) | 23 (13-33) | 25 (12-33) |
| Daily patient load, median (IQR) | 25 (16-30) | 27.5 (15-45) | 25 (16-30) |
| Consult fee in ZAR, median (IQR) | 350 (280-355) | 330 (280-250) | 350 (260-380) |

main study [33,34]. Their practices were also aptly heterogeneous to enable exploring the breadth of issues under study. GPs commonly reported seeing 'flu,' colds, fever, and headaches. Chronic issues were also common, especially among larger practices. Some described being largely involved in managing a range of health issues including geriatric conditions, arthritis, hypertension, and diabetes, but also referring complicated ones. Clients seen were generally described as of varying ages and from diverse geographical locations and communities. GPs also varied in the range of their onsite services; some offered chest X-ray and sputum sample collection services, and medicines ranging from analgesics to antipyretics and various antibiotics; others had no onsite testing or treatment supplies. Yet other GPs had 'IV' (intravenous) and 'casualty' (emergency room) equipment onsite. GPs who kept and dispensed drugs onsite seemed more prone to using empirical treatment.

In the rest of the findings, we present three interrelated themes related to general care provision by GPs. The first addresses the dynamics of antibiotic prescription, focusing on infections GPs deemed worthy of treatment with antibiotics, and their use of empirical treatment and determination of differential diagnoses. The second concerns management decisions that may be influenced by perceived or expressed pressure from clients. The final theme probes the role that economic and financial considerations play in the context of GP practices. (Table 2 summarises the key themes emerging, and Fig 1 graphically outlines the context of GPs management decisions).

### Antibiotic prescription dynamics

According to GPs, conditions viewed as complex and/or chronic (such as diabetes, HIV, hypertension, ischaemic heart disease, TB, cholesterol, trauma or ectopic pregnancy) required onward referral. These reportedly entailed expensive management regimens, with some being easier or better managed by the public health system. Facility size, onsite availability of suitable equipment, and mindfulness about private management costs for their clients were among GPs' considerations when referring chronic/ complex conditions. Conversely, conditions considered 'minor' (such as respiratory conditions indicative of bacterial infection or relating to 'flus' and colds) could be treated with broad-spectrum antibiotics at point of presentation. In settings like South Africa, the term 'flu' is often used interchangeably with influenza-like or

**Table 2. Summary of findings.**

| Main theme | Key issues |
|---|---|
| Antibiotic prescription dynamics | • Complex/chronic conditions require expensive management regimens and are better/easier managed in the public sector and thus referred. Minor conditions are treated on-site with broad-spectrum antibiotics.<br>• Determining severity and deciding on action steps is a situated, pragmatic act balancing GP's experiential insight, reading of symptoms, and patient history.<br>• Empiric treatment is common due to (i) lack of laboratory tests (ii) being used as a stopgap measure that aids diagnostic and decision-making steps; thus GPs treat and observe 'systematically', beginning with broad and/or low-strength antibiotics, and adjusting treatment, or even ordering tests or referring as needed (iii) GPs avoiding alarming patients about TB before it has been confirmed.<br>• While GPs are generally aware of recommended best practices, their described practices show contradictions, tending to weave between 'appropriate' and 'inappropriate' management actions, and drawing on a wide interpretation of guidelines. |
| Perceived client pressures | • Clients exert pressure for specific treatments; those paying cash, in particular, may expect medicines.<br>• GPs endeavor to accommodate patient expectations as they strive for compassionate, empathetic care.<br>• GPs further noted the need to help 'boost' the body's capacity to fight disease with antibiotics.<br>• Some resist, stressing to patients their role in deciding on management, and need to abet antimicrobial resistance |
| Socioeconomic considerations | • As GPs strive to serve their predominantly economically disadvantaged clients compassionately, they pursue the least burdensome management options, including trying to provide services promptly and in one place, and avoiding pharmacy trips, referrals, or return visits.<br>• GPs tend to prioritize their clientele's wellbeing over profit-making.<br>• GPs might consider TB among their differential diagnoses by factoring in patient's socioeconomic profile, including physical appearance (dressing and grooming), presenting history, and medical insurance status. |

cold-like symptoms, with little distinction between the two [50]. Thus, GPs would routinely mention 'common flu' and 'colds' in one breath. GPs described assessing presenting signs and symptoms to determine a course of action and the timing of treatment initiation. Determining symptom severity, making differential diagnoses, and choosing what action to take were situationally grounded, pragmatic, and reliant on a GP's balancing of experiential insight, symptoms reading, and client history.

*"With infections, as an experienced doctor, you know this injection doesn't work; this one works. … If a patient has a flu, you don't give them [an antibiotic]… body pain you might give [an] anti-inflammatory. However, if they've got a bad infection in the chest, you give them an antibiotic …."* (Male, aged in the middle-50s)

*"If I suspect viral [infection] … I'll probably give something just to clear the symptoms…the cough …the nasal congestion … kill the sore throat … but if I think there is secondary bacterial infection -- (for viruses, antibiotics don't work) -- I give antibiotics (**which ones?**) Penicillin … works all the time … some people react, but I will inquire about that first."* (Male, aged 30 years)

Without laboratory test results, GPs recognised the uncertainty underpinning their medical decisions. In light of this, they report giving antibiotics to avoid burdening clients with multiple trips, to eliminate potential risk of transmission to social networks (particularly where their clients generally lived in crowded conditions), or to prevent worsened outcomes from conditions that, while not yet confirmed, might just fully or partially resolve with antibiotic treatment.

*"Again… we don't have resources to test for [bacterial] infection [on site] … but I bet you this patient is going home … fifteen people in one house… chances of them getting contact with any bacteria [the patient might have] are very high… are you going to say, 'Let me take your sputum now and test if you truly have an infection, and you come back [for results]?' … it's not going to work …."* (Male, aged 55 years)

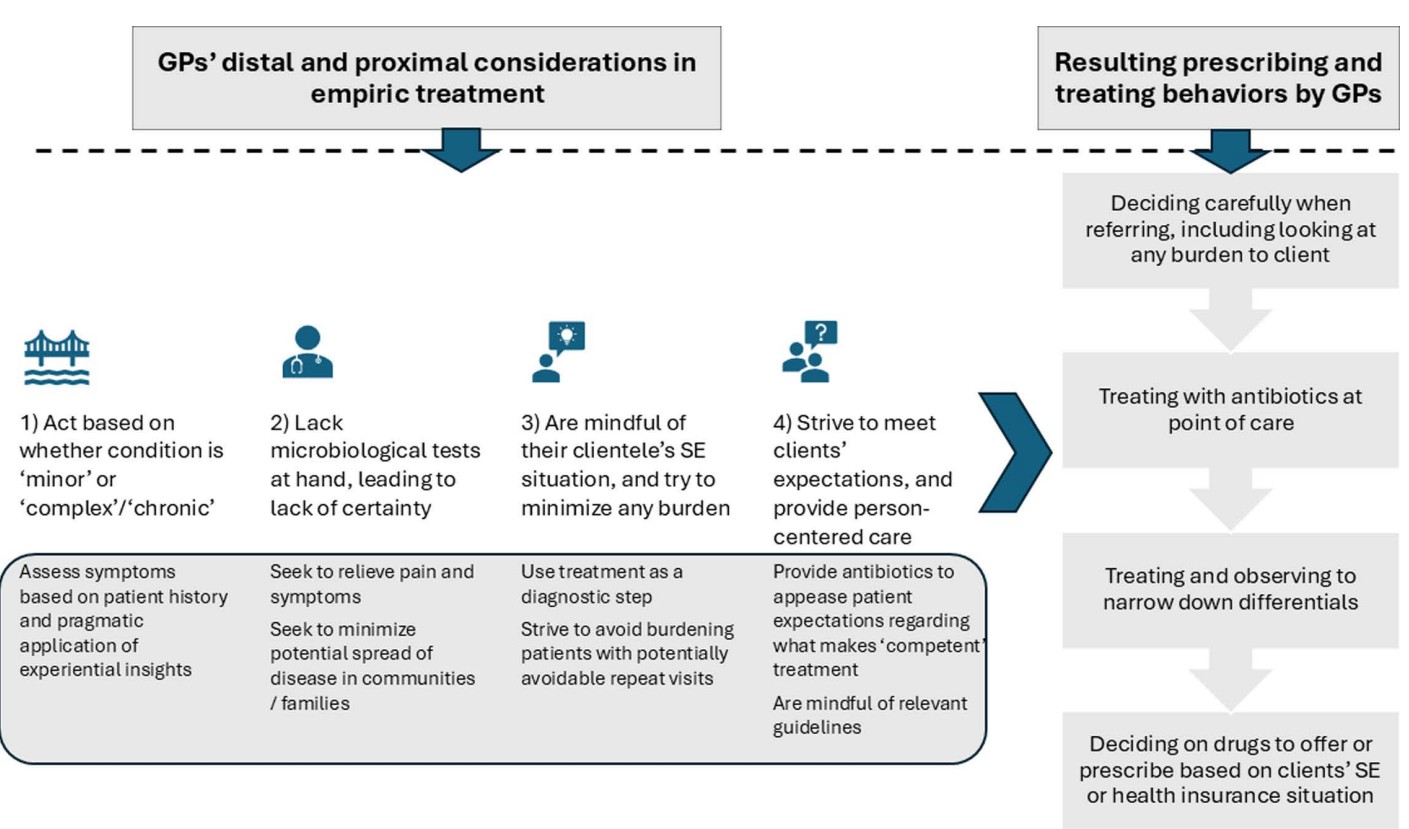

**Fig 1. Summary of findings here.**

*"Instead of giving them a script to a pharmacy … most cash [paying patients] wouldn't be happy about it … I mean the sooner you think about the GP practices: you cannot confirm with each patient exactly what sort of bacteria… we don't do microbiology … you can't do sputum and swabs on everyone; you sort of use your discretion … and cover in broad [terms] what you [think you] can."* (Male, aged 50s)

Empiric treatment was commonly viewed and employed as a stopgap measure that aided diagnostic and decision-making steps. The GPs' indicated that empirical treatment including antibiotics was widespread. Treating and observing was seen as part of a systematic-looking approach, typically beginning with broad or low strength/dose antibiotics, then increasing potency or narrowing microbial targeting as perceived necessary, and ultimately ordering investigative tests or referring if needed. During this process, progress was reviewed over periods ranging anywhere between three days and four weeks.

*"Mostly you work systematically… best is to treat the basic things... cover broadly … use an antibiotic injection and give them some oral antibiotic… So, say I was treating for … bronchitis, if you are not improving after 5 days, I've given you a strong antibiotic, I've given you an injection, the next thing let's do a sputum, let us do an X-ray... always do the basic things, then advance or refer after."* (Female, aged 30s)

Sometimes, however, GPs empirically treated with strong antibiotics *'because when people [eventually] come [in]… they have exhausted the [over-the-counter] things they [can] get from the pharmacy … [and] that obviously hasn't*

*worked…"* (Female, aged 50s). Since our work was nested within a TB-focused study, we probed GPs' considerations about TB. GPs highlighted the need for prudence and for treating and observing rather than alarming clients with a differential diagnosis that could only be confirmed with the additional step of testing. Building on the view that empirical treatment was 'systematic', GPs reiterated they rely on thorough examinations and history-taking for management decisions. Some described deferring consideration of TB for weeks until symptoms persisted or empirical treatment failed to elicit a positive response. Yet others mentioned how relatives can contribute to history-taking through clarifying symptoms and illness history.

> *"[If] their mark is suggesting bacterial... obviously they need an antibiotic, so we give them… usually something penicillin-based …and obviously if it's not improving -- you know, once we hit the two-week sort of barrier -- then you worry about TB."* (Female, aged mid-30s)

> *"Most come saying, 'I'm losing weight'. They won't say 'I'm coughing and have been sick for a while' … Until the wife, son, or mother declares 'No… it's been like this for 3 months now.' Right, then we examine them, and say, 'Listen, this is most likely TB. It could also be a normal chest infection. We give you a 3-day course and see what happens in 3 days.'"* (Male, aged 60s)

In principle, GPs seemed aware of recommended best practices. However, they revealed notable contradictions when describing their practices, sometimes weaving between 'appropriate' and 'inappropriate' management actions. One (male, age 50s) related, for instance, that he 'tested' prior to giving any medicines but proceeded to say, if not yet suspecting TB, he might give '*treatment for bronchitis, and if … not getting better … send to another differentiating diagnosis used to exclude TB*'. He later said he considered any prior treatment received by the client elsewhere, alongside presenting symptoms. He then reiterated a commonly mentioned approach, that '*just looking at a patient… (one) can tell something is not right',* although he also stated in the same interview that a person could have TB and look 'normal'.

More GPs described treating before investigating -- regardless of whether they considered TB on the basis of symptoms -- and reviewing after a few (maybe three) days for '*more definitive symptoms.*' An effusion or severe deterioration would then prompt immediate referral for chest X-ray and other tests (illustratively, two male GPs, one aged 60 years and another in his 40s, expressed these sentiments). Treating and reviewing was said to be necessary '*before worrying about TB*' because TB was considered to be '*the big one*' (female, age middle 30s). GPs also treated other urgent symptoms first (e.g., 'tight chest' or bronchitis), even in the presence of typical TB-related symptoms, as they assumed the client had, by this stage, already developed pneumonia.

Notably, while GPs might consider TB from the outset, they nevertheless drew on a wide interpretation of the national and international guidelines for antibiotic use in making decisions around and investigating for TB.

> *"With TB, you'll suspect from the [outset]. They'll come to you and say 'I've used everything… but now it's been 3 weeks or … a month. … I'm losing weight, … appetite... feeling weak. Now I am starting to cough blood, I'm having night sweats,' and you're thinking,* **Okay I think I should proceed [as TB]** *… But the protocol is at least I should give antibiotics for a week, any simple infection from pneumonia, bronchitis and everything should clear on good antibiotics. 'After a week come back so that we assess you again.' If still unwell, from there, I must do a [referral] letter."* (Male, aged 30s)

### Perceived client pressures

Clients were frequently portrayed as exerting pressure on GPs to be managed in certain ways. A common sentiment was that, because they paid for the consult, they had certain expectations, including to receive medicines. Providers noted they should try to accommodate the expectations, especially for cash-paying clients, and, to the extent possible, minimize

difficulties and inconveniences to clients while seeking healthcare. GPs also indicated that some clients were reluctant to undergo TB investigations, even after empirical treatment without improvement. One (male, age 45--55 years) illustrated how he may try to convince a client to *'wait first, 3, 4 weeks for improvement [after first treating with antibiotics] … if none and you're on medical aid (health insurance), go for TB X-rays. [But] some you truly must convince them that they must do this'*. Other GPs recounted how some clients would declare themselves as having negligible risk for TB ('*I am wealthy, I don't get TB*').

The GPs indicated they resisted the pressures in several ways. One was to clarify upfront their role '*to treat the patient to the best of [my] knowledge and based on the patient's real needs*' (male, age late 50s). They could also pretend to yield and cooperate by prescribing benign, "non-harmful" medicines. Alternatively they restressed nonpharmaceutical measures, for instance, advising clients that merely talking to the doctor was a form of treatment or intervention; or, for illnesses such as flu, recommending rest, vitamin supplementation, nasal decongesting, and analgesics to manage headaches and fever. GPs could also remind clients about risks in unnecessary use of medicines, even harnessing the authoritative stance expected of them to emphasize the message.

> "*…speak to them strongly that we don't want to cause resistance,because that's a big thing in our country…. They may not like it, but you've got to be firm … 'next time you're sick the injection won't work if you take it for every flu and small thing.'*" *(*Female, aged 30s)

In isolated instances, GPs indicated they yielded, noting the need to meet clients' expectations or to help 'boost' the body's capacity to fight disease.

> "*A lot of my patients expect injections … antibiotics … so they get [them]. … It is patients' expectations… we give in to it.*" (Male, aged 50s)

> "*Most people love injections, especially the elderly. [I commonly give] flu medications… pain injection... just boost them up … umm antibiotics like the broad-spectrum ones have severe side effects, like they need an IV … I'll just give them the pure based; I will give them penicillin … [I give antibiotics for] ear, throat and nose infections, the chest infections; umm bowel infections not as much… umm urinary tract infections, skin infection … ones I know will not resolve by themselves …. the body needs a bit of help. [It] must do the most part, but antibiotics help a little bit.*" (Male, aged 30s)

## Socioeconomic considerations

Accounts indicated that finance-related matters influenced client management. GPs saw diverse clients, many of whom were economically disadvantaged, and they described feeling obliged to serve them compassionately. Hence, socioeconomic factors were a key consideration in where to refer for tests or further management, or which drugs to prescribe. Moreover, as indicated earlier, chronic or complex conditions were regarded as expensive to manage, particularly in the private sector, a factor GPs considered when making referral decisions for uninsured clients. Some mentioned they avoided turning away clients who were short of money lest a serious event ensued; rather, they treated them 'at cost' (supposedly without any profit), or they elected to carry the losses associated with dispensing medicines below what these had cost them. GPs viewed stocking and dispensing medicines onsite as helping to minimise patient inconvenience.

Several GPs advised clients to seek (further) attention at public facilities, recognising this would be their most financially viable option since public care is free for those earning below a certain level or without income. However, balancing their goodwill against financial implications for their practice was not without tension.

> "*In general, we try to give our patients the facility of getting everything here and save them the trouble of getting medicines from the chemist… There are some who do not always have transport … we tend to be compassionate towards the patient.*" (Male, aged 60s)

*"Look, the medication part is probably why I am not rich, because I give them a lot of medication… and I'm losing a lot of money [on it]. My belief is: if you need it, even if you pay the R200 [USD15.00] consultation fee and the medication actually cost R300 [USD23.00], as long as you will be well [you get it]…."* (Male, age 30s)

Amidst these tensions, it was clear that while GPs hoped for financial profit, they also aimed to minimise the burden for the clients. Even though GPs would routinely refer insured clients to private laboratories and specialists, give them prescriptions to private pharmacies, and recommend pricier brand-name drugs, their actions did not appear to incur out-of-pocket expenses to the user. Rather, they seemed to capitalise on steps they knew were covered by health insurance programs. Giving prescriptions rather than directly dispensing drugs, for instance, allowed insured clients the agency of choosing their preferred or affordable medicines brands at the pharmacy, or, when using cash, when to procure them. When assessing TB risk.

GPs may also infer about or consider TB among differential diagnosis based on their impression of the client's socio-economic circumstances (through looking. for example, at the state of grooming and dress, tpresenting history, or medical insurance status).

*"**(You mentioned symptoms; which do you look for**?) …. coughing, white sputum... if it's TB with cavities, blood. Then cough of more than 3 weeks, losing weight, losing appetite, having night sweats, and have TB contacts. …. then obviously risk factors like HIV etc. … Poor socioeconomic status, uhm, living in shacks you know, so ja, then you are exposed."* (Male, aged 40s)

## Discussion

### Summary of the findings and comparison with global evidence

This study described the dynamics of antibiotic treatment for symptoms that may be related to TB by private GPs. In general, GPs acknowledged that treating respiratory symptoms with broad-spectrum antibiotics was a common practice. The closer look into GP decisions accorded by this study reveals how TB diagnoses are potentially delayed in private GP settings. This may be due to perceptions about clients and their expectations, and norms that deem empiric antibiotic treatment an acceptable and accessible way to bypass a lack of point-of-care tests for differential bacteriologic diagnoses including TB. Indeed, from the providers' perspective, empirical antibiotic treatment seemed to constitute a systematic way of moving towards firm diagnoses, appropriate further investigations, and ultimately reduce the perceived burden of illness and the healthcare-seeking process on clients.

We have reported in a related analysis from the parent study that GPs in this setting know how TB typically presents [33]. In the current study, however, some expressed concerns about other possible infections requiring immediate treatment. For example, symptoms that indicate "bad chest infections," "secondary bacterial infections," and infections that may spread to other members of the household appeared to warrant immediate antibiotic treatment. TB symptoms (prolonged cough, chest pain, weakness or fatigue, unintended weight loss, fever, night sweats) [51] are generally nonspecific and overlap with those of other common bacterial respiratory infections, including *S. pneumoniae*, *S. aureus* and *H. influenzae* [52–54]. In settings such as South Africa, a definitive TB diagnosis is further complicated by high HIV prevalence [1]. Moreover, the high HIV prevalence coupled with the economic marginality of their clientele populations may warrant that GPs be vigilant regarding other common infections that could result in severe complications [55–57].

It is perhaps because of this context that GPs commonly reported using antibiotics as a means to rule out other differential diagnoses. Some referred to this as a systematic way to "treat basic things" before moving on to consider sputum tests and chest X-ray if there was no improvement. This sentiment seems to explain findings in the parent study involving standardised patient visits, where many GPs considered TB at the outset but asked clients to return to probe TB further if antibiotic treatment did not resolve the symptoms [33].

A seemingly grey area for GPs is the point at which antibiotics should be introduced for respiratory symptoms -- which could be TB-related -- and how (and when) the antibiotics' effectiveness or failure to draw a response should be determined. The 2020 South African Primary Healthcare Guidelines [58] recommend the use of amoxicillin or penicillin for suspected pneumonia in adults and children while also recommending simultaneous testing by Xpert Ultra to rule out TB disease in adults. In children, further investigation for TB is recommended in instances of severe pneumonia requiring hospitalisation, and where there is history of known TB contact, malnourishment or loss of weight, and when living with HIV [59].

As several GPs reiterated, they do not have the resources on hand to initiate tests for TB or other bacterial infections, which means they send clients to private laboratories at a cost or refer to the public sector, which generally takes more time to navigate, for more in-depth evaluation. Thus, some even provided basic medications at low or no cost to the client, contrary to observations reported in other settings [60] that private providers commonly pursue profit-making through dispensing medicines. The practising environment in South Africa, as in settings such as the United States, further requires practitioners to hold licences to dispense medicines, and drug prices are fixed, leaving little room for physicians to profit from dispensing [61]. For at least some GPs, it seems that treatment with antibiotics is the simplest way to avoid undue hardships and prevent possible complications for clients.

Yet, there are a number of problems with this approach. This not only can result in extended treatment delay and worsening health but can also be influenced by healthcare inequities. For some GPs, a client with health insurance may undergo more thorough examinations, including private sector tests, because they can afford them. In contrast, there may be additional pressure on GPs to provide a "one stop shop" for those paying cash. Even if the effects are unintentional, deferring investigations into symptoms highly suggestive of TB in favour of promptly dispensing antibiotics may increase client costs associated with transport, time, and the purchase of multiple medicines. In addition to the potential to mask TB symptoms, this practice can result in further health deterioration and severe and more advanced disease, and contribute to ongoing community transmission [62–65]. The practice of profiling TB risk based on economic status can further stigmatise people diagnosed with TB; yet, given the epidemiology of TB, and increasing call to meet the needs of all affected communities including those marginalised in various ways, this may be valid grounds for medical decision-making and not a form of race or class-based profiling. Finally, GPs' intent to rule out other potential infections, in this context of high HIV prevalence might also overlook the increasing use of antiretroviral treatment and growing rates of viral load suppression, which has reshaped the profile of infections in these populations. The TB prevalence survey in South Africa reported that 39.7% of people with TB were HIV-negative while 41.9% were HIV-positive [66]. The WHO also estimates that of the 1.25 million deaths due to TB in 2023, 1.09 million were among HIV-negative people [1].

Amid diagnostic resource constraints facing private general practice, GPs also appeared to grapple with applying their professional and experiential knowledge to process clinical presentations in ways that cater to users' concerns and financial situation in these poorer communities. One result of these motley considerations is that GPs may give antibiotics, even as they may know the risk entailed [67,68]. Relatedly, some GPs also admitted to yielding to overt or implied client demands and expectations to receive antibiotics. Of note, a recent study of antibiotic use in South Africa reported that private providers worried that clients would not return if they did not receive antibiotics [69]; however, other studies noted how providers may overestimate client expectations to receive medications [70–72].

The perceived convenience of antibiotics for GPs, their reliance on experiential acumen, and their use of antibiotic treatment as a stopgap to simplify care-seeking for clients at times negated the GPs' commitment to client wellbeing and their own perceptions that they based decisions on clinical presentation and thorough history-taking. Effectively, rather than focusing on the common symptoms recommended nationally and globally (prolonged cough, chest pain, weakness or fatigue, unintended weight loss, fever, night sweats), GPs seem triggered to investigate TB only after antibiotic treatment does not relieve symptoms. Some might even trigger investigations in instances of visibly deteriorated health state and/or prolonged duration of symptoms, in direct contrast with national and international guidelines.

This study thus presents a picture where GPs and their practices (and management decisions for TB-related symptoms) are located in a complex milieu that potentiates contradictions, tensions, inconsistencies, and circumvention of norms. Thus, GPs find themselves having to navigate complex and evolving national and international guidelines for TB and other respiratory illnesses, possibilities of alternative diagnoses, inaccessible microbiological testing, and social circumstances of their clientele [73]. For some GPs, this results in deferring TB testing, placing clients (in this instance, predominantly from marginalised circumstances) at greater risk of severe TB outcomes. Our findings mirror those of a study from India, which identified key drivers of empirical therapy, including the common use of medications as diagnostic tools, the aim to provide rapid symptom relief and manage illness costs, uncertainty about TB presentation, the impact of broad-spectrum antibiotics on TB symptoms, and doubts about the accuracy of available TB tests [65].

## Conclusion

A large proportion of people globally and in South Africa use private healthcare at different stages, including for TB-suggestive symptoms [20–23]. Studies of South Africa's private sector have also shown a decreased likelihood of providers ordering TB tests, along with increased diagnostic delays [24–26]. Optimising the client-initiated pathway to TB diagnosis and treatment and enhancing the rational use of antibiotics in high-burden LMICs are important to lessen the disease burden and risk of drug resistance and toxicity [57,67,74]. The quantitative outcomes of the parent study demonstrated that GPs have a good understanding of TB symptomatology and appropriate referral practices, although there are gaps between their knowledge and actual practices [33]. This qualitative analysis illuminates what may underlie this 'know-do' gap, particularly in relation to the widespread empirical antibiotic treatment for symptoms suggestive of TB. Antibiotics are offered ahead of TB testing to treat or rule out some common infections, offer rapid symptom relief, and often as a means to prevent hardships from costly laboratory visits or the inconvenience of public sector testing for their (predominantly economically marginalised) clients. Some GPs may use it as a way to appease clients by giving them something tangible to take away. Ultimately, empirical antibiotic treatment for presentations that may be related to TB in South Africa's private sector may best be explained by the absence of accessible and affordable TB and general bacteriologic tests initiated at the point of care. Additionally, without direct access to free GeneXpert testing, such as through the public sector, GPs are forced to refer clients to the public sector, thus asking them to restart the process, which then requires them to wait in long clinic queues for nurse evaluation; or they first attempt to mitigate costs by providing empiric antibiotic treatment for other common bacterial infections.

### Limitations

Among 212 GPs involved in the main study, we interviewed a relatively small sample. However, qualitative samples are determined based on judgment, the intended use of collected information, and the particular research method being applied [75]. Our target sample was based on the study needs (as a supplementary qualitative study), including our expectation that we would achieve saturation and sample variation of GPs [76]. We believe we achieved saturation, based on our theoretical sampling and exploration of ideas [45], and also borrowing from other works, where saturation in in-depth interviews is capable of being achieved with samples of 9–17, particularly for those with relatively homogenous study populations and narrowly defined objectives [77]. While we achieved saturation, the female: male ratio of our participants [1:5] is a potential limitation; furthermore, nuances based on gender and geography, among other factors, will require more extensive research to discern.

Interviews were also conducted during working hours, and some participants thus appeared hurried. Other participants seemed concerned that they were being audited on how they conducted their work. Research assistants, nevertheless, regularly assured participants regarding the study's purpose, and, as per the consent protocol, performed ongoing consent. These steps enabled participants who were initially not at ease to relax and speak more freely as interviews progressed. We attribute GPs' wariness likely to the limited prior research on 'quality of care' in private practice, and which specifically makes use of standardised patients and in-depth interviews, and is carried out at GPs' practices. While the

dynamics may have affected the depth of interview conversations, the steps taken, alongside measures to enhance rigor in the research process, likely mitigated the impact.

While, as noted, carrying out interviews during working hours at GPs' premises might have influenced responses, this was the time the GPs were available. Moreover, we implemented both initial and ongoing consent; additionally, even though we would have scheduled times with providers, when urgent clients arrived, the clients took priority. We also informed GPs about the approximate interview duration.

It was also clear in some interviews that GPs who did not investigate TB during the standardised patient visits in the parent study assumed they had performed per guidelines (note that individual-level performance was not disclosed to participants). Some of them would comment about what they thought might lead to some GPs performing less well. Thus it was difficult sometimes to ascertain whether, when GPs articulated decision-making processes, they were referencing other GP practices or their own. Notwithstanding these overlaps, the narratives of GPs helped to draw out a vivid and complex picture of the context in which they practice, and offer explanations as well as opportunities for intervention to help achieve improved TB outcomes. The study also appears to support an observation, from a study in South Africa, that the private sector might be lagging behind with regard to managing ART among PLHIV [78].

Throughout the research process, we employed procedures to enhance rigor as recommended for naturalistic inquiry. These included: drawing out a thick description or rich account to advance the research's dependability (or capacity to be repeated) and contextual generalizability; method and researcher triangulation; training research assistants including in probing techniques; piloting the study processes and tools; holding regular peer debriefing sessions with teams members; soliciting feedback from the GPs community on preliminary interpretation of findings; pursuing ideas as they emerged, or 'discovery and verification' [40]; and using purposive and variational sampling to obtain the breadth of dimensions [39,40,44,79].

### Implications for policy, practice and future research

As the WHO puts emphasis on achieving universal health care that is safe and of good quality, addressing inappropriate management practices will likely require action from multiple fronts. This includes considering the complex context for GPs; supporting them in their desire to optimally serve clients, which could entail highlighting the downside implications of some aspects of their practices and decision-making; heightening GPs' suspicion index for TB or the prioritisation of TB testing before empiric treatment of other diseases with similar symptoms in these contexts [33]; and enabling GPs to navigate clients' pressures and expectations and the boundaries defining the latter's involvement in their own care. Perhaps most importantly, it will require the accessibility of nationally recommended diagnostic tools for clients of private care providers. In part, this could mean connecting private practices to free tests through the public sector, as has been successfully demonstrated in pilot programmes in South Africa [80]. South Africa has long recognised partnerships between the public and private sectors as a policy objective in health, and the national treasury has a unit that provides guidance on these partnerships. It has been argued, however, that stakeholders from both sectors need more mechanisms and forums for engaging in interactions in health and that contextual factors must inform the conception and management of these partnerships [81]. Our study contributes to informing how public–private health engagement can be tailored and enhanced.

### Supporting information

**S1 Text. Semi-structured provider interview guide.**
(DOCX)

**S1 Checklist. Human Participants Research Checklist.**
(DOCX)

## Acknowledgments

The authors are grateful to the participants of this study. We would also like to thank Tsatsawani Mkhombo, Sean Jooste, Monalisa Jantjies, Linda Thumba, and Alicia North at the Human Sciences Research Council, South Africa, and Sarah Wu at McGill University Health Centre, Canada, for assisting with data collection and collation; Caroline Vadnais at McGill University Health Centre for assisting with research administration issues; the doctor associations for supporting the study in many ways.

## Author contributions

**Conceptualization:** Jeremiah Chikovore, Jody Boffa, Sizulu Moyo, Madhukar Pai, Amrita Daftary.

**Data curation:** Jeremiah Chikovore, Jody Boffa, Zimasa Gavu, Angela Salomon, Amrita Daftary.

**Formal analysis:** Jeremiah Chikovore, Jody Boffa, Zimasa Gavu, Angela Salomon, Amrita Daftary.

**Funding acquisition:** Sizulu Moyo, Madhukar Pai, Amrita Daftary.

**Investigation:** Jeremiah Chikovore, Jody Boffa, Sizulu Moyo, Zimasa Gavu, Madhukar Pai, Amrita Daftary.

**Methodology:** Jeremiah Chikovore, Jody Boffa, Sizulu Moyo, Zimasa Gavu, Madhukar Pai, Amrita Daftary.

**Project administration:** Jody Boffa, Angela Salomon, Amrita Daftary.

**Resources:** Madhukar Pai, Amrita Daftary.

**Supervision:** Jeremiah Chikovore, Jody Boffa, Sizulu Moyo, Angela Salomon, Amrita Daftary.

**Validation:** Jody Boffa, Amrita Daftary.

**Writing – original draft:** Jeremiah Chikovore.

**Writing – review & editing:** Jody Boffa, Sizulu Moyo, Angela Mak, Zimasa Gavu, Angela Salomon, Madhukar Pai, Amrita Daftary.

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
