## [Decision Letter · Decision Letter 0]

28 Nov 2024

PGPH-D-24-02250

Provider perspectives on empirical antibiotic treatment for tuberculosis-like symptoms in South Africa’s private general practice sector: A qualitative study in two cities

Dear Dr. Chikovore

Thank you for submitting your manuscript to PLOS Global Public Health. After careful consideration, we feel that it has merit but does not fully meet PLOS Global Public Health’s publication criteria as it currently stands. Therefore, we invite you to submit a revised version of the manuscript that addresses the points raised during the review process.

We look forward to receiving your revised manuscript.

Kind regards,

Mohammad Shahidul Islam, PhD

Academic Editor

2. Please provide separate figure files in .tif or .eps format.

Additional Editor Comments:

Appropriate screening for tuberculosis (TB) before prescribing antibiotics for TB-like symptoms is crucial to curbing antimicrobial resistance (AMR). Such screening also helps prevent the spread of TB, as delayed treatment in suspected cases can result in transmission.

However, various factors drive healthcare providers to prescribe antibiotics without proper diagnosis. Understanding these motivations can optimize rational antibiotic use and shorten the gap between symptom onset and TB diagnosis. In this context, the study by Chikovore et al. on provider perspectives regarding empirical antibiotic treatment for TB-like symptoms in South Africa is both important and timely.

The study highlights that practitioners often prescribe antibiotics before TB testing, even when TB-like symptoms are present, to address common infections, provide rapid symptom relief, or mitigate the financial burden of costly laboratory visits for patients. These findings offer valuable insights for policymakers in high TB burden countries, such as South Africa, to design strategies that discourage unnecessary antibiotic use in private practice and advance antibiotic stewardship programs.

However, the reviewers have raised some concerns while evaluating the manuscript. Authors need to submit a revised version addressing these points before a final decision is made regarding publication in PLOS Global Health.

Reviewer #1: First of all, thank you for submitting this important manuscript on elucidating the provider perspective on empirical antibiotic treatment for tuberculosis-like symptoms. The timing of this manuscript is pivotal due to complexities and trade off the health provider needed to make to serve their patients while maintaining objectivity to the guideline. I have spotted the places where comments are required to addressed in the form of the comment. However, some of the comments I understand will require mentioning here.

A. In the abstract, you have talked about your participant selection criteria on 15 per site. I am not clear about the rationale of doing so. Even though they are two urban cities, it is imperative to have some of the attributes regarding the type of patients they have seen, their specialities. Age is an important parameter need more elaborate the diversification of the age group to understand the perception of young and the old healthcare providers.

B. You have given a glimpse on the management decision regarding the expectation of referral or the demand regarding on the spot treatment. Can you elaborate on which type of disease seen by these groups of GPs faced more demand from the patient regarding on the spot treatment.

C. Please provide some info on the multi-drug resistant TB being different and relevant to the South African context from other countries of TB. Since the paper is on TB-like symptoms, please kindly provide the data on the prevalence of TB-like symptoms and the causative pathogen that can be linked to these symptoms.

D. In line 70, please add information of the TB diagnostic coverage to understand the context better

E. The introduction also require some justification on the focusing on the need of private healthcare infrastructure. What are the failures of the public health infrastructure. It is unfortunately not clearly articulated. Regarding the data provided for the private healthcare services, how much it is attributable to the infectious diseases and how much to chronic non-communicable disease.

F. Please kindly highlight the gaps and impact of empirical antibiotic usage and why it remains a large scope of a problem in the South African landscape in terms of drug resistance, supply chain of antibiotics regarding TB-like symptoms.

G. In your participant enrolment, I have seen a higher dominance of male in both site. Can you explain a bit how much it is similar in terms of gender parity in private healthcare providers (Male-to-Female Ratio). If it is widely different, why selecting 75% male participants.

H. In the results, you have talked about the key insights with respect to providing quotes. Can you give a brief how common was each of the scenarios you have identified. Even though its a qualitative study, a distribution of the themes can be illustrated. The best can be described in the results with a brief summary of the themes from the 30 participants which will give a certain importance of the findings identified through this study. As it currently, these insights are unique and only talked about by the individual healthcare provider.

I. The scenarios and insights that has been depicted are plausible. However, they may be skewed to particular disease profile of the patients. Can you stratify the trade off on the antibiotic prescribing by patient profile which will be helpful to understand the insights better.

J. Furthermore, the insights you have shared can you differentiate on the characteristics you have depicted regarding location, years in practice (young vs old), higher patient load and consultation fees to understand the impact on the decision making regarding empirical treatment.

K. Can you highlight the scopes and threats of indiscriminate prescribing with respect to your findings.

L. Please kindly highlight in the discussion in lights of the results with regards to HIV problem. How much patients with HIV are being seens in these private practices.

Reviewer #2: This is an insightful study on how private general practitioners in South Africa navigate diagnostic challenges and socioeconomic pressures when managing TB-like symptoms, often relying on empirical antibiotics that delay TB diagnosis. Drawing on qualitative insights from 30 practitioners, it identifies key barriers and offers actionable recommendations to improve TB care and antibiotic stewardship. I commend the authors for their nuanced approach to this critical public health issue and for providing meaningful solutions to enhance TB management in resource-limited, high-burden settings. I would like to suggest some points to improve the manuscript further:

Abstract: It would be helpful to highlight the most impactful results (e.g., how empirical antibiotic treatment delays TB diagnoses) I suggest that the authors emphasise actionable recommendations, such as integrating affordable TB tests in private practices. To improve conciseness, I suggest reducing the length by removing less critical background details (e.g., TB burden statistics).

Introduction: Please consider streamlining the introduction section. The introduction is comprehensive but could be streamlined for example shifting specific details (e.g., diagnostic statistics) to the discussion or methods section. For the statistics, only include essential data to set the context. Please consider highlighting empirical antibiotic use as the issue being addressed. Introduce the research gap upfront instead of elaborating extensively on TB burden or healthcare system shortcomings. The authors could consider presenting the study's aim in a single, concise sentence with a focus on its novelty and contribution.

Methods:

1) Design & participants – Please specify any criteria for GP selection beyond variation in age, sex, and practice location. For example, were there thresholds for years of practice or experience in treating TB/HIV cases? While 30 participants were chosen to achieve thematic saturation, provide a more explicit justification or reference similar studies with comparable sample sizes.

2) Data collection – Please elaborate on steps taken to mitigate potential interviewer bias, particularly since participants were aware of being part of a TB study. I would be wise to include some explanation (potentially in the discussion section) of how conducting interviews during GP working hours might have influenced responses and suggest alternatives for future studies.

3) Data analysis – Please expand on how the coding framework was developed, particularly for the "large chunks" of data. Was it based solely on objectives, or were external coding schemes used? Authors may want to provide a more detailed explanation of how confirmatory or alternate perspectives from the research team influenced theme refinement. Please address any potential concerns about the "audit perception" by participants and how this was mitigated. For ethical purposes, authors may want to provide more specifics (i.e. strategies) on how participant confidentiality was ensured beyond using anonymized labels for quotes.

4) Rigour of the study- Please explain strategies to ensure rigour of the study (i.e. principles by Lincoln & Guba)

Results: Please include a discussion on the skewed gender distribution (25 males, 5 females) and its potential impact on findings. To improve readability, please consider organising the findings under clearly defined subheadings. For example 1) antibiotic prescription practices; 2) perceived client pressure; 3) socioeconomic considerations. If this is within the limit, consider including figures or tables for a summary of themes or decision-making factors among GPs for better readability. I suggest shortening participant quotes where possible and ensuring they directly support the narrative.

Discussion: The discussion should focus on integrating study findings with existing knowledge, addressing implications for policy and practice, and identifying avenues for further research. Please consider my comments for the discussion section as mentioned above. I would suggest the authors consider adding subheadings on these aspects:

1) Key summary of the findings

2) Strengths & limitations of the study

3) Comparison of study findings to global evidence

4) Implications for policy, practice & future research.

---

## [Decision Letter · Decision Letter 1]

17 Apr 2025

PGPH-D-24-02250R1

Provider perspectives on empirical antibiotic treatment for tuberculosis-like symptoms in South Africa’s private general practice sector: A qualitative study in two cities

Dear Dr. Chikovore,

Thank you for submitting your manuscript to PLOS Global Public Health. After careful consideration, we feel that it has merit but does not fully meet PLOS Global Public Health’s publication criteria as it currently stands. Therefore, we invite you to submit a revised version of the manuscript that addresses the points raised during the review process.

I am issuing this decision on the basis of a single reviewer's assessment to prevent further delay. Could you please consider their comments and revise your manuscript to address the comments raised?

We look forward to receiving your revised manuscript.

Kind regards,

Sarah Jose, Ph.D.

Staff Editor

Journal Requirements:

Additional Editor Comments (if provided):

Reviewers' comments:

Reviewer's Responses to Questions

**Comments to the Author**

1. If the authors have adequately addressed your comments raised in a previous round of review and you feel that this manuscript is now acceptable for publication, you may indicate that here to bypass the “Comments to the Author” section, enter your conflict of interest statement in the “Confidential to Editor” section, and submit your "Accept" recommendation.

Reviewer #2: (No Response)

2. Does this manuscript meet PLOS Global Public Health’s publication criteria ? Is the manuscript technically sound, and do the data support the conclusions? The manuscript must describe methodologically and ethically rigorous research with conclusions that are appropriately drawn based on the data presented.

Reviewer #2: Yes

3. Has the statistical analysis been performed appropriately and rigorously?

Reviewer #2: Yes

4. Have the authors made all data underlying the findings in their manuscript fully available (please refer to the Data Availability Statement at the start of the manuscript PDF file)?

Reviewer #2: Yes

5. Is the manuscript presented in an intelligible fashion and written in standard English?

Reviewer #2: Yes

6. Review Comments to the Author

Reviewer #2: Thank you for the opportunity to review this manuscript. I have a few minor comments:

1. Table 1 effectively presents provider characteristics; however, the categorisation of "Approximates age" could be more structured and presented in a more orderly manner for improved clarity.

2. While the gender imbalance is mentioned in the Discussion, its potential impact on findings has not been critically analysed.

7. PLOS authors have the option to publish the peer review history of their article (what does this mean? ). If published, this will include your full peer review and any attached files.

**Do you want your identity to be public for this peer review?** For information about this choice, including consent withdrawal, please see our Privacy Policy .

Reviewer #2: No

---

## [Decision Letter · Decision Letter 2]

16 May 2025

Provider perspectives on empirical antibiotic treatment for tuberculosis-like symptoms in South Africa’s private general practice sector: A qualitative study in two cities

PGPH-D-24-02250R2

Dear Dr. Chikovore,

We are pleased to inform you that your manuscript 'Provider perspectives on empirical antibiotic treatment for tuberculosis-like symptoms in South Africa’s private general practice sector: A qualitative study in two cities' has been provisionally accepted for publication in PLOS Global Public Health.

Best regards,

Julia Robinson

Executive Editor

Reviewer Comments (if any, and for reference):

Reviewer's Responses to Questions

**Comments to the Author**

1. If the authors have adequately addressed your comments raised in a previous round of review and you feel that this manuscript is now acceptable for publication, you may indicate that here to bypass the “Comments to the Author” section, enter your conflict of interest statement in the “Confidential to Editor” section, and submit your "Accept" recommendation.

Reviewer #2: All comments have been addressed

2. Does this manuscript meet PLOS Global Public Health’s publication criteria ? Is the manuscript technically sound, and do the data support the conclusions? The manuscript must describe methodologically and ethically rigorous research with conclusions that are appropriately drawn based on the data presented.

Reviewer #2: Yes

3. Has the statistical analysis been performed appropriately and rigorously?

Reviewer #2: N/A

4. Have the authors made all data underlying the findings in their manuscript fully available (please refer to the Data Availability Statement at the start of the manuscript PDF file)?

Reviewer #2: Yes

5. Is the manuscript presented in an intelligible fashion and written in standard English?

Reviewer #2: Yes

6. Review Comments to the Author

Reviewer #2: Thank you for addressing the comments thoroughly.

7. PLOS authors have the option to publish the peer review history of their article (what does this mean? ). If published, this will include your full peer review and any attached files.

**Do you want your identity to be public for this peer review?** For information about this choice, including consent withdrawal, please see our Privacy Policy .

Reviewer #2: No
